# A frequentist one-step model for a simple network meta-analysis of time-to-event data in presence of an effect modifier

Matthieu Faron[1,2]*, Pierre Blanchard[1,3], Laureen Ribassin-Majed[1,4], Jean-Pierre Pignon[1,4], Stefan Michiels[1,4], Gwénaël Le Teuff[1,4]

1 Oncostat U1018, Inserm, Université Paris-Saclay, Équipe Labellisée Ligue Contre le Cancer, Villejuif, France, 2 Service de chirurgie viscérale oncologique, Gustave Roussy, Villejuif, France, 3 Service de radiothérapie, Gustave Roussy, Villejuif, France, 4 Service de Biostatistique et d'Épidémiologie, Gustave Roussy, Université Paris-Saclay, Villejuif, France

* matthieu.faron@gustaveroussy.fr

## Abstract

### Introduction

Individual patient data (IPD) present particular advantages in network meta-analysis (NMA) because interactions may lead an aggregated data (AD)-based model to wrong a treatment effect (TE) estimation. However, fewer works have been conducted for IPD with time-to-event contrary to binary outcomes. We aimed to develop a general frequentist one-step model for evaluating TE in the presence of interaction in a three-node NMA for time-to-event data.

### Methods

One-step, frequentist, IPD-based Cox and Poisson generalized linear mixed models were proposed. We simulated a three-node network with or without a closed loop with (1) no interaction, (2) covariate-treatment interaction, and (3) covariate distribution heterogeneity and covariate-treatment interaction. These models were applied to the NMA (Meta-analyses of Chemotherapy in Head and Neck Cancer [MACH-NC] and Radiotherapy in Carcinomas of Head and Neck [MARCH]), which compared the addition of chemotherapy or modified radiotherapy (mRT) to loco-regional treatment with two direct comparisons. AD-based (contrast and meta-regression) models were used as reference.

### Results

In the simulated study, no IPD models failed to converge. IPD-based models performed well in all scenarios and configurations with small bias. There were few variations across different scenarios. In contrast, AD-based models performed well when there were no interactions, but demonstrated some bias when interaction existed and a larger one when the modifier was not distributed evenly. While meta-regression performed better than contrast-based only, it demonstrated a large variability in estimated TE. In the real data example, Cox and Poisson IPD-based models gave similar estimations of the model parameters.

**Data Availability Statement:** All simulation codes are available on GitHub https://github.com/Oncostat/One_step_frequentitst_IPD_NMA.

**Funding:** The author(s) received no specific funding for this work.

**Competing interests:** The authors have declared that no competing interests exist.

Interaction decomposition permitted by IPD explained the ecological bias observed in the meta-regression.

## Conclusion

The proposed general one-step frequentist Cox and Poisson models had small bias in the evaluation of a three-node network with interactions. They performed as well or better than AD-based models and should also be undertaken whenever possible.

## 1 Introduction

Direct treatment effect (TE) estimation between two treatments from a single randomized clinical trial (RCT) is prone to sampling variability and the "traditional" meta-analyses (MA) aim to reduce this variability by gathering TE estimates from several trials. Network meta-analyses (NMA), first described in 2002 [1] extend this principle when more than two therapeutic options are available. NMA can also be applied to interventions such as diagnostic or preventive measures. In NMAs of treatments, a network is created whose nodes are therapeutic options (subsequently designated by "treatment") and edges (subsequently designated by "comparisons") are pairwise treatment comparisons. The information used for TE estimation can vary from one edge to another, from no comparison to several ones. According to the network geometry, treatment comparison might be estimated by either indirect comparison only or combining information from both direct and indirect comparison [2].

In both MA and NMAs, the first step is to identify all studies (published or unpublished) to reduce publication bias. This step can be long and usually relies on standardized search strategies [3]. While, any type of study can be used (observationaland RCT), the quality of the MA/NMA relies on the quality of the underlying studies. We focused on RCTs as they are considered to provide the the highest level of evidence for treatment effect evaluation in the context of evidence-based medicine and because the incorporation of non RCT in NMA would address new challenges not considered here [4]. For each RCT, an assessment of the risk of bias should be made. The trial TE can be retrieved in two ways, based on either AD or IPD. In AD, the most common, researchers only retrieve the TE and its precision for each study. This approach is easier and faster to perform because researchers "only" have to retrieve the information published in the literature (article, abstract, . . .). In IPD, researchers retrieve the full dataset of each trial, including covariates for each patient, and the TE, along with its precision, is recalculated. The latter is a fastidious work because of the difficulty to effectively retrieve individual information and laws regarding IPD are becoming more complex. Yet, IPD is considered the gold standard as it allows more control on the data to update follow-up and evaluate the heterogeneity of TE by including covariate-treatment interactions [5]. For NMA based on non-randomized studies, IPD may also be used to reduce confusion bias [6].

In the absence of a TE modifier (i.e., interaction), both AD-MA and IPD-MA should lead to similar TE estimation [7]. In the presence of a modifier, AD-MA may lead to incorrect TE estimation [7]. Meta-regression might be an alternative to identify variation in TE related to trial aggregated patient characteristics, but is less powerful and prone to ecological bias [7]. Besides, sufficient information on the TE modifiers may not have been reported in the original publication (or even recorded). It should be noted that the TE modifier can also be at the trial-level (rather than at the patient-level) and in this case, it may be taken into account in AD-MA. IPD-MA is able to estimate both between- and within-trial interaction and leads to more

accurate TE estimation [8]. Interactions are even more challenging in NMA as they may only exist in some comparisons and not in others and the distribution of the modifiers may not be the same across comparisons. If not taken into consideration, these differences can sometimes invalidate the estimation of TE, as illustrated by Jansen and Naci in 2013 [9]. Besides, these differences can, in the absence of IPD, lead to difficulties in verifying the statistical assumption of NMA: transitivity [10–12] and consistency [13]. Transitivity corresponds to the fact that all patients of the network could have been randomized to any treatment options or in other words that there are no systematic differences between the comparisons. Transitivity assumption does not hold when an effect modifier has not a similar distribution across comparisons. Consistency, which can be considered as the manifestation of transitivity in the data, means that for a particular treatment comparison where both direct and indirect evidence are available, they provide similar estimations [14].

Over the past years, this interaction topic in NMA has been the subject of several works for a binomial outcome [15]. However, due to more complex models, potential issues for time-to-event outcomes have been minimally studied. Three options are mainly available for NMA with survival data: (i) AD with its limitations with a variety of methods [2], some of them allowing meta-regression [16]; (ii) when IPD is available, a two-step approach, in which pair-wise treatment comparisons are independently evaluated in a first step and then combined in a second step, but continuous variables need to be categorized; and (iii) an IPD one-step model that often relies on a hierarchical random effects model and, most of the time, is implemented in a Bayesian framework for computational reasons [17].

The aim of this study was to develop a general one-step IPD approach for a time-to-event outcome in a three-node network under the frequentist paradigm. The bias in indirect and mixed TE estimation in presence of a TE modifier measured at patient level was evaluated. The usual AD methods with their expected bias were used as references. A simulation study was used to precisely quantify the bias and a real example of an IPD-NMA from the Meta-analyses of chemotherapy in Head and Neck Cancer (MACH-NC) and Radiotherapy in Carcinomas of Head and Neck (MARCH) was used for illustration [18,19]. This NMA was chosen because of a previously observed interaction between age and treatment effect as published in the main papers of MARCH/MACH-NC.

## 2 Statistical approaches for a network meta-analysis

### 2.1 IPD-based model

In the one-step IPD-based approach, a multilevel hierarchical model was used in which patients were nested within trials and trials nested within a comparison. For a time-to-event outcome, the TE was the log(Hazard Ratio) (log[HR]). In a three-treatment network (A-B-C), the three evaluated treatments A, B and C were coded as two dummy variables. Only two coefficients $\beta_{AC}$ and $\beta_{BC}$ representing the TEs of the comparisons A versus C (A-C) and B versus C (B-C) were needed, as the third one $\beta_{AB}$ for the comparison A versus B (A-B), also called a *functional parameter*, could be indirectly deduced by contrast under the consistency assumption from the following equation:

$$\log(HR_{AB}) = \beta_{AB} = \beta_{AC} - \beta_{BC}$$

Similarly, the covariate-treatment interaction for A-B comparison $\gamma_{AB}$ could be estimated from the two covariate-treatment interactions from A-C comparison $\gamma_{AC}$ and B-C comparison $\gamma_{BC}$ as:

$$\gamma_{AB} = \gamma_{AC} - \gamma_{BC}$$

Therefore, for a given age x, the log hazard-ratio of TE comparing A to B is given by:

$$\beta_{AB}^{age=x} = (\beta_{AC} - \beta_{BC}) + (\gamma_{AC} - \gamma_{BC}) \times x$$

To ensure that the variance is correctly estimated in every group, treatment indicators were coded as -0.5/+0.5 and not 0/1 [20]. The sign of the coefficients were arbitrarily set to follow the direction of the arrow in Fig 1 and respect the consistency. A summary of the coding is provided in Table 1.

**2.1.1. Mixed-effects Cox's proportional hazard model (*IPD-CoxME*).** A multilevel Cox proportional hazard model (CoxME) can be used for IPD NMA [21,22]. However, this approach leads to extensive calculation times when the complexity of the network and the number of random effects increase. BIn order to reduce computation time in the simulation study, this model was not used in the simulation study, but only in the MARCH/MACH-NC NMA data application.

**2.1.2. Mixed-effects Poisson's model (*IPD-Poisson1*).** A mixed-effects Poisson model (Poisson ME1) can be used, after transformation of IPD, to perform survival analysis and has led to results similar to the Cox model and has already been used for standard meta-analyses [23,24]. The transformation consisted in dividing the individual follow-up time into several time intervals. Baseline risk was coded as a global intercept plus a trial-specific random effect. Baseline risk was allowed to vary in each of the intervals coded with a spline to allow non-linear relation while maintaining continuity between interval-specific risks [23]. Between-trial heterogeneity of the TEs was taken into account by a random effect. Therefore, for patient $i$, in trial $j$, comparison $k$, and time interval $l$, the model is written as follows:

$$d_{ijkl} \sim Poisson(\mu_{ijkl})$$

where $d_{ijkl}$ corresponds to the event indicator taking the value of 0 or 1, and

$$\log(\mu_{ijkl}) = (\beta_0 + b_{1k} + b_{0j}) + spline(\lambda_l) + \log(y_{ijkl}) +$$
$$\beta_{age} \times age_{ijk} + (\beta_1 + b_{2j}) \times treat1_{ijk} + (\beta_2 + b_{3j}) \times treat2_{ijk}$$
$$+ \gamma_1 \times treat1_{ijk} \times age_{ijk} \times \gamma_2 \times treat2_{ijk} \times age_{ijk}$$

where $\beta_0$ is the overall baseline risk, $b_{1k}$ represents the deviation in the k-th comparison from the overall underlying baseline risk, $b_{0j}$ represents the deviation in the $j$ trial from the comparison specific baseline risk and accounts for the second level of clustering (trials nested into comparison), $spline(\lambda_l)$ is the spline function modeling the baseline risk over time-interval, the constant $\log(y_{ijkl})$ is the time at risk of patient considered as an offset in the model, $\beta_{age}$ is age-coefficient regression, $age_{ijk}$ is the patient age, $\beta_1 + b_{2j}$ is the random A-B TE with the average log hazard ratio $\beta_1$ plus a trial-specific random effect $b_{2j}$, $\beta_2 + b_{3j}$ is the random A-C TE with the average log hazard ratio $\beta_2$ plus a trial-specific random effect $b_{3j}$, $\gamma_1$ is the age-treatment A-C interaction effect and $\gamma_2$ is the age-treatment A-B interaction effect. The different between-trial random effects were assumed to follow a multivariate normal distribution with mean zero and a variance-covariance matrix defined as:

$$\begin{pmatrix} b_{0j} \\ b_{1k} \\ b_{2j} \\ b_{3j} \end{pmatrix} \sim MVN \left( \begin{bmatrix} 0 \\ 0 \\ 0 \\ 0 \end{bmatrix} \begin{bmatrix} \delta_1^2 & 0 & 0 & 0 \\ 0 & \delta_2^2 & 0 & 0 \\ 0 & 0 & \tau_1^2 & 0 \\ 0 & 0 & 0 & \tau_2^2 \end{bmatrix} \right)$$

The variances $\delta_1^2$, $\delta_2^2$ respectively measured the between-comparison and between-trial

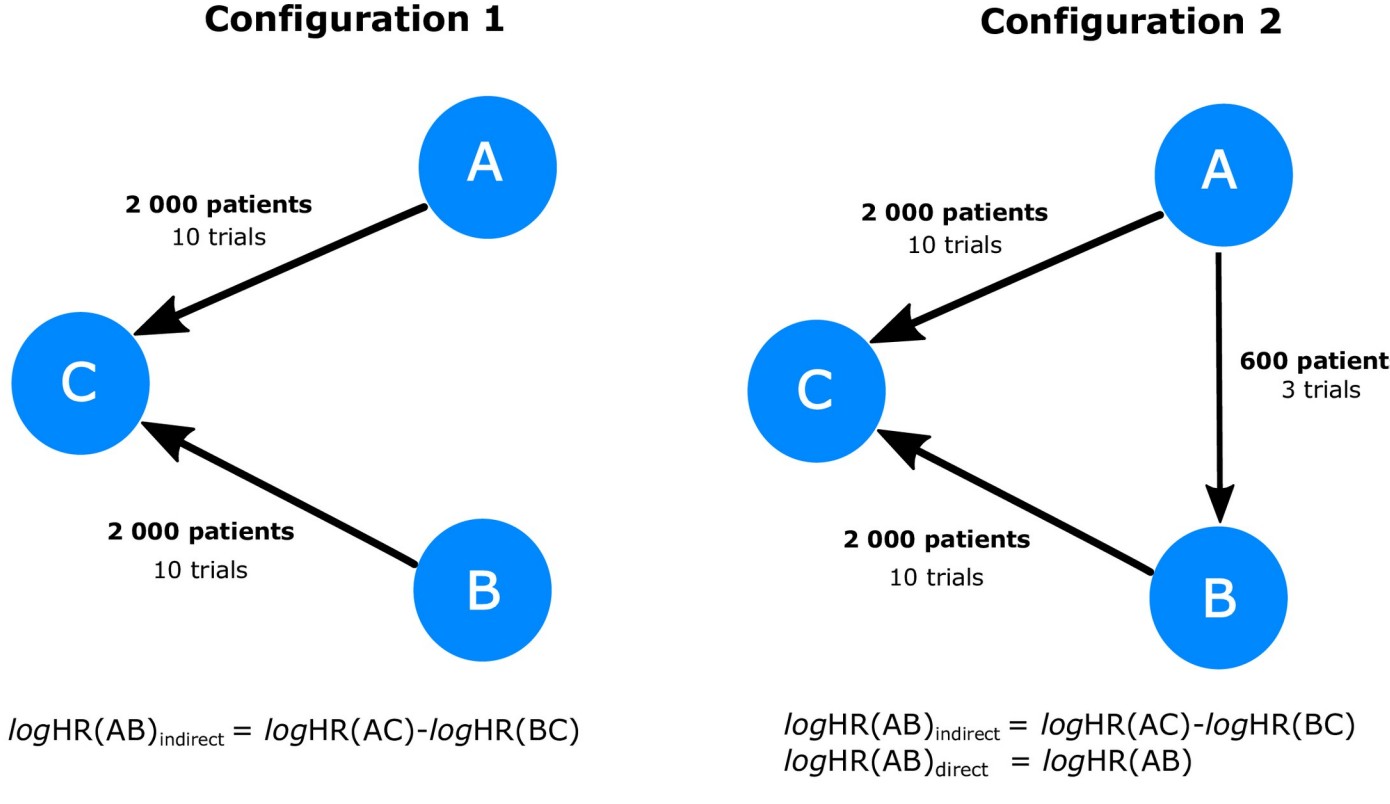

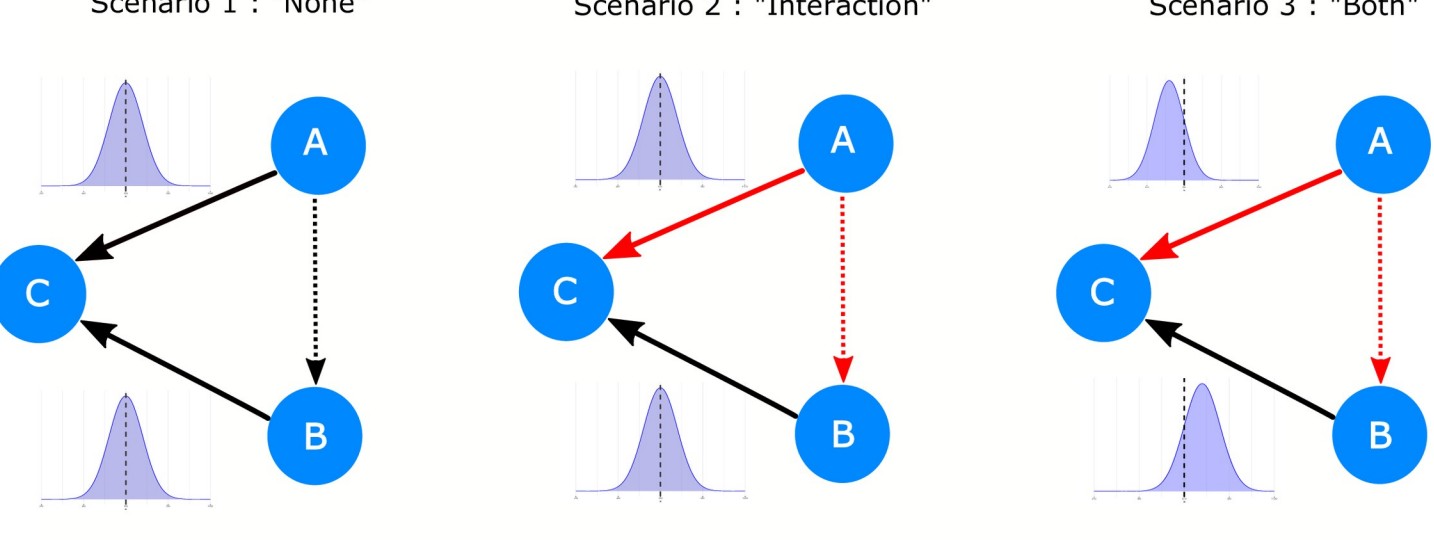

**Fig 1. Three-node network according to available trials, the distribution of age and presence of age-treatment interaction.** Upper panel represents the two configurations, and lower panel represents the three scenarios. For each configuration, the three scenarios were simulated. The total number of patients and trial are indicated over each arrow. The direction of the arrows indicates the direction (sign) of the comparison when estimating the treatment effect; therefore, X → Y indicates a X versus Y comparison and $log(HR)(XY)$ denotes the log hazard ratio of the X versus Y comparison. Density plots are the distribution of age with the dashed line indicating age 60. For scenarios 1 and 2, the distribution of age was identical for A-C and B-C pairwise treatment comparisons, while the distribution is opposite in scenario 3: Mean age is higher in A-C trials and lower in B-C trials. Each configuration was simulated according to the three scenarios as represented in the lower panel.

Solid arrows indicate trials available in all configurations and dashed arrows indicate trials only available in configuration 2. Black arrows indicate no age-treatment interaction, while red arrows indicate an age-treatment interaction.

heterogeneity in the baseline risk, and $\tau_1^2$, $\tau_2^2$ measured the between-trial heterogeneity in the TE A-B and A-C. For simplicity, we assumed independent random effects for intercepts and slopes. Multi-arm studies can be used with proper coding.

As specified earlier, only two dummy variables for treatment effects were necessary in network with or without AB trials, but when AB trials are present, patients must be carefully coded (Table 1).

**2.1.3. Decomposition of the interaction (*IPD-Poisson2*).** Both Cox and Poisson models can be reparametrized to separate the global interaction into a within- and between-trial interaction [8]. In the previous model, the interaction between treatment indicator 1 and age was parameterized as:

$$\gamma_1 \times treat1_{ij} \times age_{ij}$$

The model can be reparametrized as follows:

$$\gamma_{1w} \times treat1_{ij} \times (age_{ij} - \overline{age_j}) + \gamma_{1b} \times treat1_{ij} \times \overline{age_j}$$

where $\gamma_{1w}$ is the within-trial, $\gamma_{1b}$ is the between-trial interaction terms and $\overline{age_j}$ is the mean age of patients in trial *j*. The difference between $\gamma_{1w}$ and $\gamma_{1b}$ is, therefore, an indicator of an ecological bias. This decomposition was applied to both interaction terms (i.e., the two treatment indicators). We used this parameterization of the Poisson model (PoissonME2) only for the NMA of the real data as in the simulation study, the same mean age was simulated for all trials within a comparison (no ecological bias).

Both the IPD-Poisson1 and IPD-Poisson2 models can use age as a continuous variable or categorical one with dummy coding.

## 2.2 AD-based model

Models used for AD-NMA were based on trial-level TE (comparison) and varied on how they combined these TEs. These models are mainly used when only aggregated data are available.

**Table 1. Treatment coding in the one-step individual patient data (IPD) model and the aggregated data (AD) meta-regression.**

| Type of trial | Treatment allocated | Treatment 1 Dummy variable | Treatment 2 Dummy variable* |
|---|---|:---:|:---:|
| *IPD-Model* | | | |
| A vs C | A | +0.5 | 0 |
| | C | -0.5 | 0 |
| B vs C | B | 0 | +0.5 |
| | C | 0 | -0.5 |
| A vs B | A | +0.5 | -0.5 |
| | B | -0.5 | +0.5 |
| *AD-Meta-regression* | | | |
| A vs C | - | 1 | 0 |
| B vs C | - | 0 | 1 |
| A vs B | - | 1 | -1 |

*Treatment-2 dummy variables is necessary in meta-regression for a network where three types of trials (AB, AC and BC) are present.

The IPD-based model includes a random intercept for the trial-specific baseline risk and two random effects (one for each dummy variable) for the trial-specific TEs.

They can also be used with IPD in a two-step approach. At the first step, adjusted or unadjusted TEs are computed for each trial. At the second step they are combined with these models.

**2.2.1. Contrast-based model (*AD-Netmeta*).** Recently, a frequentist AD-NMA method was proposed by Rücker et al. [25] This graph-based method based on the analogy with electrical network allowed consideration of various situations, including multi-arm studies, inconsistency and random effects. It has been shown to be consistent with other approaches while being computationally simple. This method does not currently allow the inclusion of treatment modifiers, and therefore, will be limited to the situation in which the interaction is not taken into consideration. For $j$ two-arms trial of n treatments, the model is written as follows:

$$y = X\mu + b + \varepsilon$$

with $y$ as a vector of $j$ values of log(HR), $X$ as a ($j$, n-1) design matrix of the model, $\mu$ as a vector of underlying means of the (n-1) TE, $b$ as a vector of $j$ trial-specific random effects, and $\varepsilon$ as a vector of $j$ random error with:

$$b \sim N(0, \tau^2)$$

$$\varepsilon \sim N(0, V); V = diag\{v_j\}$$

with $V$ as a diagonal matrix whose $j^{th}$ entry is $v_j$, the variance of the $j^{th}$ trial.

For example, in a three-treatment network with three two-arm studies: two A-C trials, one B-C trial and no A-B trial, the model would be:

$$\begin{bmatrix} y_1 \\ y_2 \\ y_3 \end{bmatrix} = \begin{bmatrix} 1 & 0 \\ 1 & 0 \\ 0 & 1 \end{bmatrix} \begin{bmatrix} \mu_{AC} \\ \mu_{BC} \end{bmatrix} + \begin{bmatrix} b_1 \\ b_2 \\ b_3 \end{bmatrix} + \begin{bmatrix} \varepsilon_1 \\ \varepsilon_2 \\ \varepsilon_3 \end{bmatrix}$$

with indirect comparison AB estimated from $\mu_{AB} = \mu_{AC} - \mu_{BC}$.

The design matrix and $V$ can be adapted to allow for multi-arm studies.

In an IPD-two step approach, this model can be used to derive adjusted TE. Continuous variables are transformed to categorical ones then an adjusted TE for each category is computed separately in each trial. Then, TEs are combined separately for each category.

**2.2.2. Meta-regression (*AD-Metareg*).** Under the consistency hypothesis, a NMA can also be seen as a special case of meta-regression using *n-1* dummy variables to code the *n* treatment comparison [2]. This method has the advantages to allow the introduction of between-trial interaction terms. A disadvantage is that multi-arm trials are harder to account for. For a three treatment (A, B, C) NMAs with A-C and B-C trials, the model is written for trial $j$ as follows:

$$y_j = (\beta_0 + b_{0j}) + \beta_1 \times trial_j^{AC} + \beta_2 \times \overline{age}_j + \gamma_1 \times \overline{age}_j \times trial_j^{AC} + e_j$$
$$b_{0j} \sim N(0, \tau^2)$$

$$e_j \sim N(0, v_j)$$

where $y_j$ is the log(HR) of treatment vs. reference in trial $j$, $trial_j^{AC}$ is a treatment indicator with value 1 for an A-C trial and 0 for a B-C trial [2] (Table 1), $\overline{age}_j$ is the mean age in trial $j$, $v_j$ is the variance of the log(HR) in trial $j$ in order to "weight" the trial by its inverse variance and $b0_j$ is a random variable of the TE with the between-trial heterogeneity $\tau^2$.

The same TEs and modifiers, as in the IPD models of section 2.1, can be computed as a function of the model parameters:

$$\beta_{AC} = \beta_0 + \beta_1$$

$$\beta_{BC} = \beta_0$$

$$\gamma_{AC} = \beta_2 + \gamma_1$$

$$\gamma_{BC} = \beta_2$$

Under the consistency assumption, indirect TE of the comparisons A-B and its interaction with age (i.e., the functional parameters) are obtained by:

$$\beta_{AB} = \beta_{AC} - \beta_{BC}$$

$$\gamma_{AB} = \gamma_{AC} - \gamma_{BC}$$

In a closed network i.e. when A-B trials exist, a second treatment indicator, along with a second interaction term, is needed to account for the three trial possibilities (Table 1).

This model can be used with a 2-class categorical variable but will have results hard to interpret for a variable with more categories.

## 3 Simulation study

Most of the work is a simulation study with no real data. All patients in the real data applications signed an informed consent in their respective trial.

We performed a simulation study to evaluate the bias and precision of the estimation of indirect and mixed treatment comparisons in presence of modifier effect in the framework of IPD NMA for time-to-event data. Simulations were based on a three-node network (A, B and C). Event times of each two-arm randomized trial comparing two of the three treatments (A, B and C) with balanced (1:1) allocation were simulated using a Weibull distribution (intercept = 8, scale = 0.73). These two Weibull parameters led to a survival of 65% and 41% at 5 and 8-years, respectively, combined with a variation in shape according to randomly assigned age and treatment group [26]. Censoring times were drawn from Weibull distribution with intercept = 8.5 and scale = 0.35. All observations were administratively censored after seven years (two years of accrual period and five years of follow-up). With this setting, the censoring rate in each trial was 27%. The follow-up time was divided in l = 7 time-intervals of one-year for mixed effect Poisson models. These different parameters for simulating time-to-event data were set to mimic the timeframe of overall survival in an oncology trial for localized solid tumors. The TE was identical for the A-C and B-C pairwise treatment comparisons. Under the consistency assumption, this led to a null TE in the A-B pairwise treatment comparison. The coefficients associated to treatment effect were set to $\log(0.82) = -0.2$ and $\log(0.61) = -0.5$ with the sign of the coefficient arbitrary chosen, so that A and B were more effective than C. A between-trial heterogeneity of A-C and B-C TE was assumed through a random effect following $N(0, \tau^2)$ and a deviation from the overall underlying baseline risk through another random effect following $N(0, \delta^2)$. Two values of the variance of the random effects were considered: a large $(0.1^2)$ and a small $(0.01^2)$. A continuous variable representing the age of patients was drawn from a normal (mean = 60, standard deviation [SD] = 8) distribution and considered either as a continuous variable (IPD-Poisson and AD-Metareg) or as a categorical one with four categories: $< 55$, 55–60, 60–65, and $> 65$ (IPD-Poisson). Cut points were chosen *a priori*,

so that patients were balanced between categories. We assumed that, in every simulation, age was associated to the survival by a coefficient regression of $\log(\text{HR}) = 0.011$ for each increase of one year of age.

Two configurations of the three-node NMA without loop (configuration 1) and with loop (configuration 2) with different scenarios were considered. The configuration 1 consisted of a three-node network in which two treatments (A and B) have been separately evaluated against a common comparator C, but not directly (no A-B trial). For each comparison, we simulated 10 trials of 200 patients each (**Fig 1**). In this network, the TE comparing A and B were estimated by indirect comparison only.

The configuration 2 completed the network of the configuration 1 by simulating A-B trials generating a closed loop. We simulated three A-B trials of 200 patients each (Fig 1) with A-B TE verifying the consistency hypothesis. In this network, the TE comparing A and B was estimated by combining direct and indirect information. In order to avoid overrepresentation of the part of direct information in the mixed treatment comparison, we chose a total number of randomized patients in the A-B trial smaller than those in the A-C and B-C trials. Too much direct information would have hidden the impact of the potentially biased indirect estimation on the mixed treatment comparison.

For each configuration (Fig 1 upper panel), three scenarios were simulated according to the distribution of age and the presence of age-treatment interaction between the comparisons of the network (Fig 1).

- **Scenario 1: "*None*":** ages in A-C and B-C trials were drawn from the same distribution and no age-treatment interaction was present. Under the consistency assumption, this led to a null TE in the indirect A-B pairwise treatment comparison without interaction.

- **Scenario 2: "*Interaction*":** ages in A-C and B-C trials were drawn from the same distribution. Age-treatment interaction was simulated for A-C treatment arbitrarily chosen so that TE decreases by 25% for an increase of one SD of age (quantitative interaction). Under the consistency assumption, this led to an interaction in the A-B comparison identical to that in A-C and a null TE at age 60.

- **Scenario 3: "*Both*":** ages were drawn from different distributions in A-C and B-C trials: one SD younger in the A-C comparison and one SD older in the B-C comparison. Age-treatment interaction was simulated for A-C treatment, as in scenario 2. Under the consistency assumption, this led to a null TE at age 60 in the indirect A-B comparison with the same interaction present in A-C.

For configuration 2, which included A-B trials in a three-node network, age was drawn in the same way, in comparison to scenarios 1 and 2 for the configuration 1.

For each combination of simulation parameters and scenarios (Table 2), the mean of the TE-related regression coefficients (log HR) across 1,000 replications was calculated. For age as a *continuous variable*, we are interested in (i) the parameter associated to the TE for a patient with a mean age denoted as $TE_{\overline{age}}$ or here $TE_{age = 60}$ and called the conditional treatment effect and (ii) the parameter associated to the interaction (age x TE) which is the variation in TE (vTE) for an increase of one unit in age and denoted $vTE_{age+1}$. For age as a *categorical variable*, we are interested in the TE for each age class denoted: $TE_{<55}$, $TE_{55-60}$, $TE_{60-65}$ and $TE_{>65}$. Results are also presented as mean bias, defined as the mean of the differences between the estimated and the true value. Errors are presented as empirical standard error (ESE) and average standard error (ASE). ESE was defined as the SD of the estimated parameter over the 1,000 replicates. ASE was defined as the average of the standard errors estimated by each model [27,28].

All simulations were done with the R 3.5.2 software (The R Core Team, Vienna Austria, 2020) with the furrr 1.14.0 (Vaughan, 2018) packages for multicore computing and a L'Ecuyer-CMRG random number generator to ensure that each worker started at a different seed. Cox model was estimated by the coxme package 2.2–16, multilevel Poisson's with lme4 1.1.21 [29], contrast based-AD with Netmeta 1.1.0 and meta-regression with metaphor [30]. The R code for all simulations is available on GitHub (https://github.com/Oncostat/One_step_frequentitst_IPD_NMA).

## 4 MARCH/MACH-NC: A real example of an IPD network

The MACH-NC compared chemoradiotherapy (CTRT) to radiotherapy (RT) (first update: 64 trials and 12,129 patients) [19] and the MARCH compared mRT to RT (initial meta-analysis: 15 trials and 6,515 patients) [18]. Together, they formed a NMA without trials comparing mRT to CTRT [31]. The primary outcome was overall survival and published results from MARCH using IPD showed an age-RT-mRT interaction [32] and patients in the RT vs. CTRT MA were younger than in the RT vs. mRT MA, leading to a situation similar to our scenario 3 of the configuration 1. The AD- and IPD-based models proposed in the statistical section were applied to this three-node NMA to illustrate the potential bias in the indirect estimation of the mRT-CTRT pairwise treatment comparison. Some simplifications were made for illustrative purposes; readers interested in the clinical application are referred to the original publications (S1 text).

## 5 Simulation results

### 5.1 Convergence

No models failed to converge in the IPD one-step framework. In comparison, there were some convergence issues in the AD-meta-regression in 47 simulations (0.2%) in

**Table 2. Simulation parameters.**

| Parameters | Possible values |
|---|---|
| Treatment effect | • -0.2 (hazard ratio = 0.82)<br>• -0.5 (hazard ratio = 0.61) |
| Variance of random effect of baseline hazard ($\delta_i^2$, i = 1,2) | • $0.1^2$<br>• $0.01^2$ |
| Variance of random effect of treatment effect ($\tau_i^2$, i = 1,2) | • $0.1^2$<br>• $0.01^2$ |
| Configuration | • $1^*$: No trials comparing A vs. B<br>• 2: 3 trials of 200 patients in the direct comparison A vs. B |
| Scenario | • None (no age-treatment interaction, no distribution differences of age)<br>• Interaction (age-treatment interaction in A-C trials, no distribution differences of age)<br>• Both (age-treatment interaction in A-C trials, mean age 52 years in A vs. C trial and 68 years in B vs. C trials) |
| Total number of possibilities | 2 x 2 x 2 x 2 x 3 = 48 |
| Number of replications for each combinations of parameters | 1000 |
| Overall number of simulations: | 48 x 1000 = 48000 |

$^*$: Configuration 1 is a three-node (A-B-C) network, including A-C and B-C trials only. For each comparison, we simulated 10 trials, including 200 patients per trial (See Fig 1 upper panel).

configuration 1, and three (0.01%) in configuration 2. The network with the simplest configuration (i.e., no closed loops, configuration 1) with the simplest scenario (scenario 1: "none") confirmed the validity of our algorithm for generating individual time-to-event data for a three-arm network.

## 5.2 Directly estimated parameters

The two direct TE (A-C and B-C), along with their interactions, were estimated using PoissonME1 with no bias in all situations. Results for the simulation in which TE = -0.5 and $\sigma = \tau$ = 0.01 are given in Table 3; results for all simulations are given in S1 Table. Similarly, the age effect was also estimated with no bias in all settings (S1 and S2 Figs).

## 5.3 A-B TE estimation

Table 4 summarizes the result of the A-B TE estimation with the IPD-based Poisson 1 model when age was included as a categorical variable and the true TE was -0.5, and the between-trial heterogeneity for baseline risk and the between-trial heterogeneity of the TE were 0.01. The results of an AD-based approach in which all trials had given TE adjusted for each age classes which is similar to an IPD-two-step approach, were given as a comparison. Results for all simulations are represented in S2 Table for configuration 1 and S3 Table for configuration 2. Fig 2 shows the results of the simulation for the model when age was a continuous variable where TE = - 0.5, $\sigma$ = 0.01 and $\tau$ = 0.01. Two AD-based approaches (AD-Netmeta, which does not allow for interaction, and AD-meta-regression, which does) are given as a comparison. Results for all simulations are given in S4 Table.

**5.3.1. Indirect A-B TE estimation (configuration 1).** *Scenario 1*: *"None"*. The model estimated the A-B TE with small to no bias with age either as a continuous (Fig 2) or categorical variable (Table 4). As expected in this balanced design, AD-models (AD-Netmeta, AD-Metareg) also performed very well.

*Scenario 2*: *"Interaction"*. When age was continuous, the model correctly estimated both the conditional TE for a 60-old-year patient ($TE_{age = 60}$) with bias = 0.004 and the conditional TE for an increase of one unit in age ($vTE_{age+1}$) with bias = 0.000. As a reference, both AD-based models also correctly estimated the conditional TE (bias for AD-Netmeta = 0.006, for AD-Metareg = 0.005). The AD-Metareg (which allows for interaction) also correctly estimated the $vTE_{age+1}$ (bias = 0.004). AD-Metareg, when compared to our IPD model, exhibited a large variation in estimated interaction coefficient (ESE = 0.170, ASE = 0.032, vs ASE = ESE = 0.010 for IPD) (Fig 2).

When age was categorical, IPD-Poisson1 correctly estimated all age classes with bias lower than 0.01.

*Scenario 3*: *"Both"*. When age was continuous, the model correctly estimated the conditional TE ($TE_{age = 60}$) and its variation with age (bias = -0.003 for $TE_{age = 60}$ and bias = 0.000 for $vTE_{age+1}$) demonstrating its ability to adequately estimate unbalanced design with interactions. As expected in this complex design, both AD-based approaches demonstrated bias for the conditional TE (bias for AD-Netmeta = -0.12, for AD-Metareg = -0.041) and its variation with age for AD-Metareg (bias = -0.002). Our model was remarkably stable with ESE = 0.010 and ASE = 0.010 for $TE_{age = 60}$ and ESE = 0.113 and ASE = 0.0116 for $vTE_{age+1}$. As a comparison, AD-Meta-regression provides unstable estimations across replications resulting to a large variation in quantiles of observed values ($TE_{age = 60}$: ESE = 1.381, ASE = 2.121; $vTE_{age+1}$: ESE = 0.179, ASE = 0.033) (Fig 2).

When age was categorical, our model had small bias for all categories with a maximum bias = -0.034 for the age > 65 category) (Table 3).

**Table 3. Simulation results for the A-C and B-C comparisons when treatment effect = -0.5, the between-trial heterogeneity for baseline risk was set at 0.01 and the between-trial heterogeneity of the treatment effect set at 0.01.**

| | | Configuration 1 | | | | Configuration 2 | | | |
|---|---|---|---|---|---|---|---|---|---|
| | | A-C | | B-C | | A-C | | B-C | |
| Scenario | Parameters | Bias | ESE | Bias | ESE | Bias | ESE | Bias | ESE |
| | | Age as a continuous variable | | | | | | | |
| None | $TE_{age=60}$ | -0.001 | 0.053 | -0.001 | 0.052 | -0.002 | 0.047 | 0.001 | 0.046 |
| | $vTE_{age+1}$ | 0 | 0.007 | 0 | 0.006 | 0 | 0.006 | 0 | 0.006 |
| Interaction | $TE_{age=60}$ | 0.002 | 0.061 | -0.001 | 0.054 | 0.002 | 0.053 | 0.001 | 0.047 |
| | $vTE_{age+1}$ | 0 | 0.008 | 0 | 0.007 | 0 | 0.007 | 0 | 0.006 |
| Both | $TE_{age=60}$ | -0.002 | 0.083 | 0.001 | 0.074 | 0.001 | 0.066 | 0.002 | 0.062 |
| | $vTE_{age+1}$ | 0 | 0.008 | 0 | 0.007 | 0 | 0.006 | 0 | 0.006 |
| | | Age as a qualitative variable | | | | | | | |
| None | $TE_{<55}$ | -0.003 | 0.103 | -0.007 | 0.104 | -0.004 | 0.095 | 0.001 | 0.095 |
| | $TE_{55-60}$ | -0.001 | 0.108 | 0.006 | 0.106 | -0.006 | 0.098 | -0.003 | 0.1 |
| | $TE_{60-65}$ | -0.004 | 0.107 | -0.004 | 0.11 | 0.001 | 0.094 | 0.003 | 0.096 |
| | $TE_{>65}$ | 0.001 | 0.103 | 0 | 0.098 | 0 | 0.09 | 0.002 | 0.087 |
| Interaction | $TE_{<55}$ | 0.004 | 0.122 | -0.003 | 0.107 | 0.002 | 0.105 | 0 | 0.095 |
| | $TE_{55-60}$ | -0.003 | 0.125 | -0.003 | 0.106 | 0.001 | 0.115 | -0.004 | 0.1 |
| | $TE_{60-65}$ | 0.006 | 0.125 | 0 | 0.109 | 0 | 0.111 | 0.002 | 0.099 |
| | $TE_{>65}$ | 0.002 | 0.115 | 0 | 0.104 | 0.006 | 0.097 | 0.003 | 0.09 |
| Both | $TE_{<55}$ | -0.035 | 0.081 | -0.004 | 0.242 | -0.037 | 0.077 | -0.028 | 0.157 |
| | $TE_{55-60}$ | -0.004 | 0.135 | 0.002 | 0.169 | -0.002 | 0.12 | 0.009 | 0.137 |
| | $TE_{60-65}$ | -0.011 | 0.175 | -0.002 | 0.117 | -0.005 | 0.146 | 0 | 0.109 |
| | $TE_{>65}$ | -0.032 | 0.254 | 0.002 | 0.065 | -0.01 | 0.162 | 0 | 0.063 |

Configuration 1 is a 3-nodes model without a closed loop, and configuration 2 is the same model with a closed loop.

TE: treatment effect, vTE: Variation in TE for an increase of one unit in age, ESE: Empirical standard error.

**5.3.2. Mixed A-B TE estimation (configuration 2).** When compared to configuration 1, the results for our model were mostly unchanged (Fig 2). Yet, as a comparison, better precision was observed with lower ESE and ASE for AD-based models in scenarios "none" and "interaction". In scenario "both", adding direct comparison reduced but did not cancel the bias observed in configuration 1 in the conditional TE ($TE_{age=60}$) for AD-Netmeta (bias = -0.075 vs -0.12 in configuration 1). AD-Metareg demonstrated a smaller confidence interval, but still exhibit a large variability with ESE = 0.110 and ASE = 1.276 for $TE_{age=60}$ and with ESE = 0.135 and ASE = 0.000 for $vTE_{age+1}$.

## 6 Application to the MARCH/MACH-NC NMA

The MARCH/MACH-NC NMA has an identical geometry as that of the simulated network in configuration 1 since there is no closed loop (no trial comparing mRT and CTRT). In addition, mean ages were different in the two comparisons (59.9±10.1 vs 56.7±9.84) and an interaction with age was observed in the mRT vs RT comparison. This configuration was close to our scenario 3 "both", but with smaller age differences between comparisons.

Age was used as a continuous variable for this analysis. Table 5 and Fig 3 report the results of conditional TE for two direct comparisons (mRT vs RT and CTRT vs RT), one indirect comparison (mRT vs CTRT) and two interactions estimated from three versions of IPD-based models: mixed-effects Cox, and mixed-effects Poisson with and without decomposition of interaction. Results from two AD-based approaches were given as a comparison.

**Table 4. Simulation results of the indirect and mixed A-B treatment comparisons in the different scenarios when the true treatment effect was -0.5, the between-trial heterogeneity for baseline risk was set at 0.01 and the between-trial heterogeneity of the treatment effect was set at 0.01.**

| Scenario | Age | True value of log HR | IPD-Poisson | | | | AD-Netmeta | | | |
|---|---|---|---|---|---|---|---|---|---|---|
| | | | Mean | Bias | ESE | ASE | Mean | Bias | ESE | ASE |
| **Configuration 1: indirect comparison only** | | | | | | | | | | |
| **1: None** (no interaction. same age distribution) | TE$_{<55}$ | 0.000 | 0.004 | 0.004 | 0.143 | 0.020 | 0.000 | 0.000 | 0.076 | 0.006 |
| | TE$_{55-60}$ | 0.000 | -0.008 | -0.008 | 0.154 | 0.024 | 0.000 | 0.000 | 0.076 | 0.006 |
| | TE$_{60-65}$ | 0.000 | 0.000 | 0.000 | 0.154 | 0.024 | 0.000 | 0.000 | 0.076 | 0.006 |
| | TE$_{>65}$ | 0.000 | 0.001 | 0.001 | 0.143 | 0.020 | 0.000 | 0.000 | 0.076 | 0.006 |
| **2: Interaction** (interaction. same age distribution) | TE$_{<55}$ | -0.154 | -0.147 | 0.008 | 0.158 | 0.025 | 0.006 | 0.160 | 0.081 | 0.032 |
| | TE$_{55-60}$ | -0.038 | -0.037 | 0.000 | 0.162 | 0.026 | 0.006 | 0.044 | 0.081 | 0.008 |
| | TE$_{60-65}$ | 0.038 | 0.044 | 0.006 | 0.159 | 0.025 | 0.006 | -0.032 | 0.081 | 0.008 |
| | TE$_{>65}$ | 0.154 | 0.156 | 0.002 | 0.155 | 0.024 | 0.006 | -0.148 | 0.081 | 0.028 |
| **3: Both** (interaction. different age distribution) | TE$_{<55}$ | -0.154 | -0.186 | -0.031 | 0.259 | 0.068 | -0.120 | 0.034 | 0.084 | 0.008 |
| | TE$_{55-60}$ | -0.038 | -0.044 | -0.006 | 0.214 | 0.046 | -0.120 | -0.083 | 0.084 | 0.014 |
| | TE$_{60-65}$ | 0.038 | 0.029 | -0.009 | 0.213 | 0.045 | -0.120 | -0.158 | 0.084 | 0.032 |
| | TE$_{>65}$ | 0.154 | 0.121 | -0.034 | 0.264 | 0.071 | -0.120 | -0.275 | 0.084 | 0.082 |
| **Configuration 2: mixed (direct/indirect) comparison** | | | | | | | | | | |
| **1: None** (no interaction. same age distribution) | TE$_{<55}$ | 0.000 | -0.005 | -0.005 | 0.112 | 0.013 | -0.003 | -0.003 | 0.057 | 0.003 |
| | TE$_{55-60}$ | 0.000 | -0.003 | -0.003 | 0.125 | 0.016 | -0.003 | -0.003 | 0.057 | 0.003 |
| | TE$_{60-65}$ | 0.000 | -0.002 | -0.002 | 0.115 | 0.013 | -0.003 | -0.003 | 0.057 | 0.003 |
| | TE$_{>65}$ | 0.000 | -0.002 | -0.002 | 0.105 | 0.011 | -0.003 | -0.003 | 0.057 | 0.003 |
| **2: Interaction** (interaction. same age distribution) | TE$_{<55}$ | -0.154 | -0.153 | 0.002 | 0.129 | 0.017 | 0.003 | 0.157 | 0.064 | 0.029 |
| | TE$_{55-60}$ | -0.038 | -0.033 | 0.005 | 0.135 | 0.018 | 0.003 | 0.040 | 0.064 | 0.006 |
| | TE$_{60-65}$ | 0.038 | 0.036 | -0.002 | 0.131 | 0.017 | 0.003 | -0.035 | 0.064 | 0.005 |
| | TE$_{>65}$ | 0.154 | 0.157 | 0.003 | 0.113 | 0.013 | 0.003 | -0.152 | 0.064 | 0.027 |
| **3: Both** (interaction. different age distribution) | TE$_{<55}$ | -0.154 | -0.164 | -0.010 | 0.156 | 0.024 | -0.075 | 0.079 | 0.065 | 0.011 |
| | TE$_{55-60}$ | -0.038 | -0.049 | -0.011 | 0.153 | 0.024 | -0.075 | -0.037 | 0.065 | 0.006 |
| | TE$_{60-65}$ | 0.038 | 0.033 | -0.004 | 0.152 | 0.023 | -0.075 | -0.113 | 0.065 | 0.017 |
| | TE$_{>65}$ | 0.154 | 0.144 | -0.010 | 0.160 | 0.026 | -0.075 | -0.229 | 0.065 | 0.057 |

Age was considered as a categorical covariable with classes: < 55, 55–60, 60–65 and > 65.

Configuration 1 represents a three-arm network with B-C and A-C trials, from which we estimated the indirect treatment effect A-B. Configuration 2 represents a three-arm network with A-B, A-C and B-C trials, from which we estimated the mixed (direct and indirect) treatment effect A-B. AD: aggregated data, IPD: individual patient data, ESE: empirical standard error, ASE: average standard error.

MARCH/MACH-NC: Meta-analyses of Chemotherapy in Radiotherapy in Carcinomas of Head and Neck/Meta-analysis of Chemotherapy in Head and Neck Cancer, log(HR): log(hazard ratio), mRT: modified radiotherapy, RT: radiotherapy, CTRT: chemoradiotherapy, CoxME: mixed-effects Cox's model, Poisson ME1: mixed-effects Poisson's model with one interaction term for a particular treatment effect, Poisson ME2: mixed-effects Poisson's model within and between interaction term for a particular treatment effect (b: between-trial, w: within-trial), Netmeta: aggregate data-based model using contrasts (mRT vs RT and CTRT vs RT), Meta-regression: aggregate data-based model using contrasts (mRT vs RT and CTRT vs RT) and adjusted on age, NA: not applicable.

Estimation of the conditional TEs for the two direct comparisons (mRT-RT and CTRT-RT) were similar between models with similar uncertainty. IPD-based models without interaction decomposition (CoxME and PoissonME1) estimated a significant interaction at 5% between mRT vs RT and age (0.006 (SD = 0.003); p = 0.046) and no significant interaction between

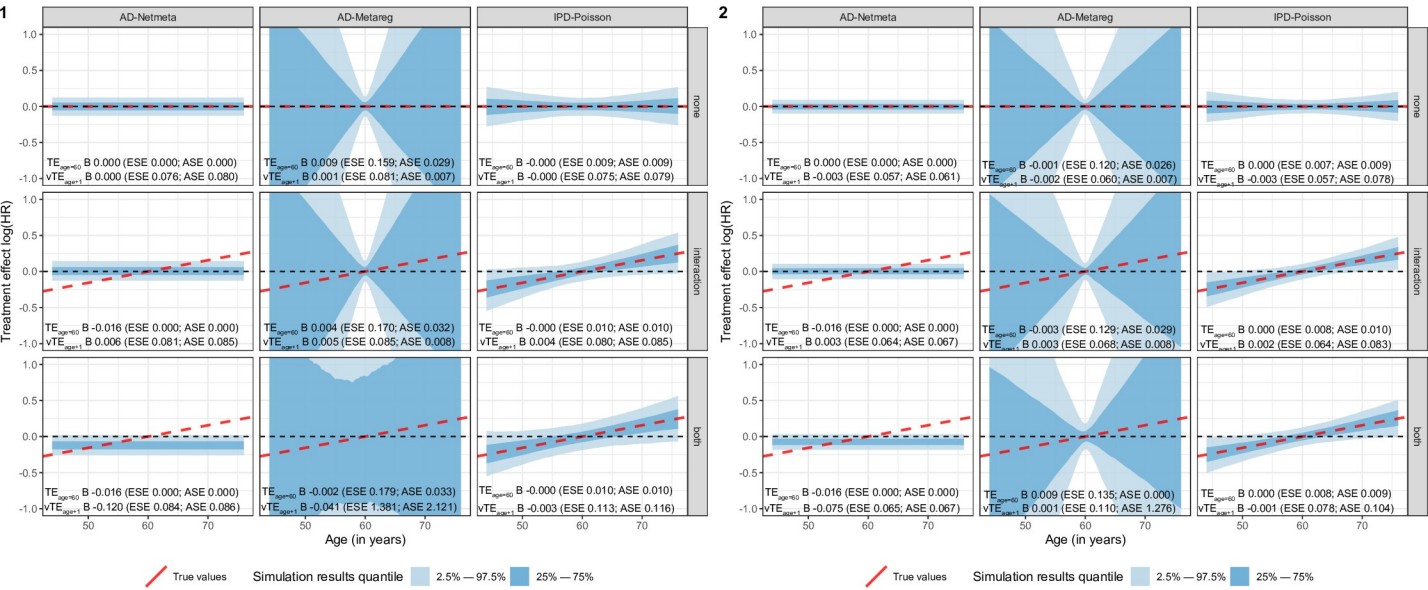

**Fig 2. Simulation results for A-B treatment comparison in different scenarios and configurations.** The indirect (configuration 1, left part) and mixed (configuration 2, right part) are represented with the three scenarios (none, interaction, and both) as rows. Age was considered a continuous variable for a treatment effect = -0.5, a between-trial heterogeneity of the baseline risk set at $\sigma = 0.01$, and a between-trial heterogeneity of the treatment effect set at $\tau_1 = \tau_2 = 0.01$. Red-dashed lines represent the true value of A-B treatment effect (log[HR]) according to age. Blue shaded zone represents the 2.5–97.5% and 25–75% quantiles of the estimated treatment effect across the 1,000 replications. IPD: individual patient data, AD: aggregated data, $TE_{age = 60}$: conditional treatment effect at age 60, the mean age of the simulation, $vTE_{age +1}$: the conditional variation in treatment effect with age, B: bias, ESE: empirical standard error, ASE: average standard error.

CTRT vs RT and age (0.002 (SD = 0.002); p = 0.317). Meta-regression estimated larger point estimates of interaction 0.032 (SD = 0.019) and 0.023 (SD = 0.010), respectively, with larger variability, as reported in Fig 3. The Poisson model with decomposition of the interaction (PoissonME2) estimated (i) a within-trial interaction terms close to those of the IPDs models and (ii) a between-trial interaction close to those of the meta-regression. This result indicated an ecological bias in the estimation of interaction terms when using the meta-regression model. For summarize, these different results suggest/show that, (i) to perfectly identify the structure of the data, the interest of interaction decomposition in an IPD-based approach should be considered, and (ii) while better than the simpler AD-based approach, meta-regression was not as effective as the IPD-based approach.

Concerning the indirect comparison (mRT-CTRT), no significant difference between mRT and CTRT was observed in the IPD-based models, as well as in the meta-regression models (all 95% confidence intervals included the value zero). Yet, a significant difference in favor of mRT was observed in the AD-Netmeta modeI, as reported by the 95% confidence interval in Fig 3 (last column). SDs were similar across the models.

These results were in accordance with those observed for the configuration 1 and the scenario 3 "both" of the simulated study (Fig 2, left panel at bottom). They illustrate the ability of one-step IPD-based models to estimate unbiased parameters, even when interaction is present and the bias magnitude of AD-based methods.

## 7 Discussion

NMA can be used to estimate TEs when there is no (configuration 1) or few (configuration 2) pairwise comparisons between two treatments and are usually approached with AD-based models or in a Bayesian framework. In this paper, we proposed a frequentist one-step IPD-

**Table 5. Analysis of the real data network meta-analysis MARCH/MACH-NC with IPD- and AD-based models.**

| Parameters | Models | Log(HR) | Standard deviation | 95% Confidence interval |
|---|---|---|---|---|
| mRT vs RT | CoxME | -0.123 | 0.045 | -0.211; -0.035 |
| (direct comparison) | Poisson ME1 | -0.120 | 0.048 | -0.215; -0.026 |
| $TE_{\overline{age}}$ | Poisson ME2 | -0.146 | 0.041 | -0.225; -0.066 |
| | Netmeta | -0.120 | 0.056 | -0.225; -0.066 |
| | Meta-regression | -0.161 | 0.058 | -0.274; -0.047 |
| CTRT vs RT | CoxME | -0.139 | 0.032 | -0.203; -0.075 |
| (direct comparison) | Poisson ME1 | -0.138 | 0.032 | -0.202; -0.075 |
| $TE_{\overline{age}}$ | Poisson ME2 | -0.107 | 0.033 | -0.173; -0.042 |
| | Netmeta | -0.148 | 0.033 | -0.212; -0.084 |
| | Meta-regression | -0.110 | 0.034 | -0.176; -0.044 |
| Interaction between | CoxME | 0.006 | 0.003 | 0.000; 0.012 |
| age (continuous) | Poisson ME1 | 0.006 | 0.003 | 0.000; 0.012 |
| and (mRT vs RT) | Poisson ME2b | 0.028 | 0.012 | 0.004; 0.053 |
| $vTE_{age+1}$ | Poisson ME2w | 0.005 | 0.003 | -0.001; 0.011 |
| | Netmeta | NA | NA | |
| | Meta-regression | 0.032 | 0.019 | -0.005; 0.069 |
| Interaction between | CoxME | 0.002 | 0.002 | -0.002; 0.007 |
| age (continuous) | Poisson ME1 | 0.002 | 0.002 | -0.002; 0.007 |
| and (CTRT vs RT) | Poisson ME2b | 0.023 | 0.010 | 0.003; 0.043 |
| $vTE_{age+1}$ | Poisson ME2w | 0.001 | 0.002 | -0.004; 0.006 |
| | Netmeta | NA | NA | |
| | Meta-regression | 0.023 | 0.010 | 0.003; 0.043 |
| mRT vs CTRT | CoxME | -0.016 | 0.078 | -0.168; 0.136 |
| (indirect comparison) | Poisson ME1 | -0.018 | 0.081 | -0.176; 0.140 |
| $TE_{\overline{age}}$ | Poisson ME2 | 0.038 | 0.074 | -0.107; 0.183 |
| | Netmeta | -0.148 | 0.065 | -0.275; -0.021 |
| | Meta-regression | 0.032 | 0.091 | -0.147; 0.211 |
| Interaction between | CoxME | -0.004 | 0.006 | -0.014; 0.007 |
| age (continuous) | Poisson ME1 | -0.004 | 0.006 | -0.015; 0.007 |
| and (mRT vs CTRT) | Poisson ME2b | -0.005 | 0.022 | -0.049; 0.039 |
| (indirect) | Poisson ME2bw | -0.004 | 0.006 | -0.015; 0.007 |
| $vTE_{age+1}$ | Netmeta | NA | | |
| | Meta-regression | -0.009 | 0.029 | -0.066; 0.048 |

In this open network, mRT vs RT and CTRT vs RT relies on direct comparison, while mRT vs CTRT relies on indirect only.

based mixed Poisson models for time-to-event outcomes. In a simulated three-node network with (configuration 2) or without (configuration 1) closed loop, our models performed well in all scenarios. Notably, even in the presence of an interaction with an unbalanced distribution of the TE modifier, the model was able to correctly estimate both the conditional TE at mean age and its variation with age. This model also demonstrated remarkable stability across simulations with low ESE. In this simulation study, AD-based models served as a reference of expected magnitude of bias. As expected, AD-based models produce similar results than IPD-based models when no interaction exists. This was not the case in the presence of an interaction in which AD-based models were unreliable in a form that depended on whether one is interested by the TE at any age or only the TE at the mean age. In the former case, in the presence of any interaction (scenario 2 and 3), TE estimates from AD-based models that did not

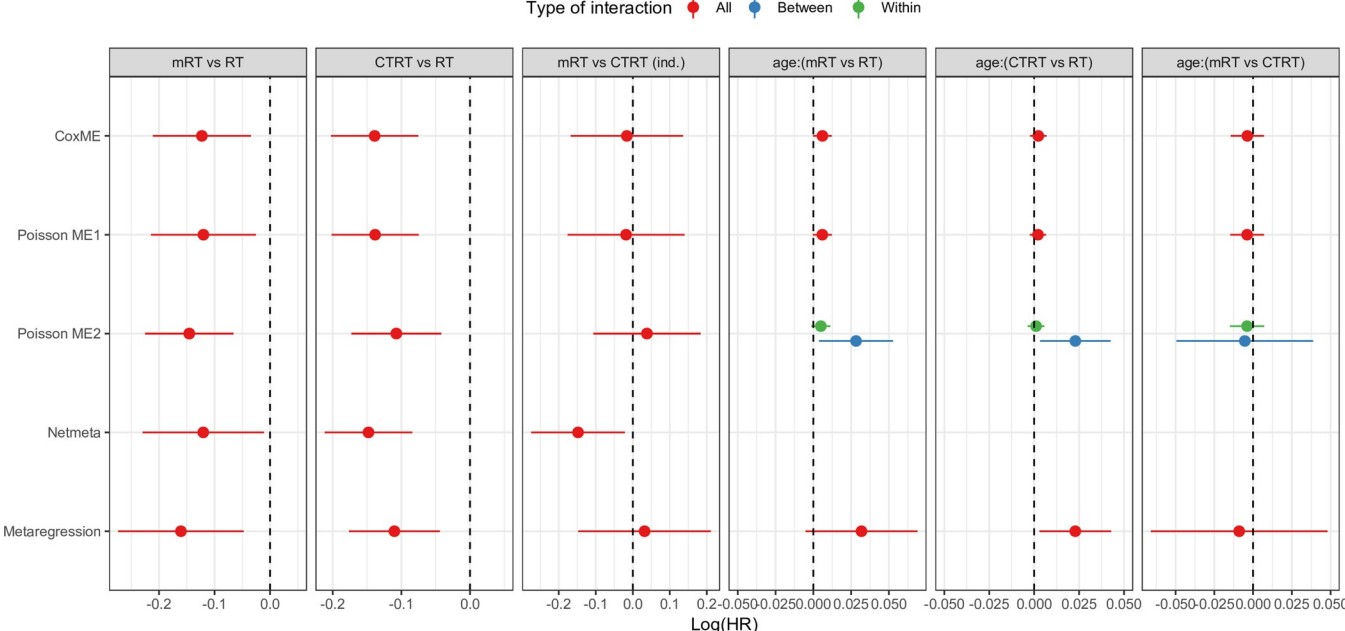

**Fig 3. Estimation of regression coefficients (log[HR]) by the models in the MARCH/MACH-NC dataset.** log(HR): log(hazard ratio), MARCH/MACH-NC: Meta-analyses of Chemotherapy in Radiotherapy in Carcinomas of Head and Neck/Meta-analysis of Chemotherapy in Head and Neck Cancer, CoxME: mixed-effects Cox Model, Poisson ME1: mixed-effects Poisson without interaction decomposition, Poisson ME2: mixed-effects Poisson with interaction decomposition, mRT: modified radiotherapy, RT: radiotherapy, CTRT: chemoradiotherapy, mRT vs RT) and age:(CTRT vs RT) are the interaction between age (entered as a continuous variable) and treatment effect, ind.: indirect estimation of the treatment effect since no trial comparing mRT to CTRT exist. Vertical dashed lines indicate no effect. Blue and green bars indicate the between- and within-trial decomposition of the interaction, respectively.

allow for interaction (Netmeta) were logically biased. AD-based models that allowed for interaction (meta-regression) were less biased, but demonstrated large ASE and ESE over 1,000 replicates, suggesting that in a single experiment, results may be unreliable, particularly when there are few studies. For the latter case, conditional TE at mean age may be biased or not in presence of an interaction according to the distribution of the modifiers. In scenario 2 (in which the modifier was evenly distributed), conditional TEs at mean age were not biased. In scenario 3 (in which the modifier did not have the same distribution in all comparisons), the conditional TE at mean age was biased in AD-based models.

Similar findings were observed on real data. One-step IPD-based Cox and Poisson models led to remarkably similar estimates and found no statistically significant differences between the indirectly compared treatments. On the other hand, an AD-based model, ignoring age, would have concluded that mRT was superior to chemoradiotherapy at a 0.05 significance level. Meta-regression performed well in this (highly powered) datasets and correctly estimated the TE with a non-significant difference. Yet, this dataset illustrated an advantage of the one-step model and an issue of meta-regression: the ecological bias that led to the overestimation of the interaction magnitude. Decomposition of the interaction in IPD-based models, as proposed by Hua et al. [8] revealed that the within-trial interaction (estimated by standard IPD models) was much lower than the between-trial interaction (estimated by meta-regression).

Our study also demonstrates the possibility to use either the Cox or Poisson model in a frequentist paradigm, both of them leading to similar results with our dataset. All models were successfully applied to the real data, including 79 trials and more than 18,000 patients, without convergence issues. IPD-based models were not too computationally intensive and took, on a standard desktop computer, 400 seconds for the mixed-effects Cox model and 100 seconds for

the mixed-effects Poisson model. These figures are reasonable for a single application, notably when compared to Bayesian approaches. The Cox model however can become very computer intensive as compared to the Poisson implementation, and for this reason, we only used the Poisson model in the simulation. We used here the Poisson model as an alternative to the Cox model for survival analysis but other teams use the flexible Royston-Parmar model [20].

Correctly estimating networks may be computationally challenging. Both Bayesian and frequentist estimation tools are available for AD-NMA and considered to deliver similar results. For IPD, most teams use either frequentist two-step or Bayesian one-step IPD. The former use IPD data to derive trial-specific log hazard ratios adjusted for other variables which are then combined. The latter allows more complex models, sometimes at the price of long calculation times. In practice, very few one-step frequentist studies have been published, especially for time-to-event outcomes, as specifying the model may be challenging and models with multiple random effects and interactions may never converge. The approach used here was to replace the traditional Cox model with a Poisson generalized mixed linear model, which often benefits of more robust software implementations. In this way, Ollier et al. proposed to extend this model by a penalized Lasso Poisson model for more complex NMA allowing to include covariates effects, inconsistency terms, covariate-treatment interactions and non-proportional treatment effects [33]. Another approach was proposed by Jackson et al. for models in MA for binary outcomes [34] and could be adapted to the IPD-NMA to solve convergence issues. In this approach, the model is gradually simplified (i.e., by removing random effects or interaction terms) until convergence is obtained. Others have proposed to combine RCT and non RCT study in the network. Cameron et al. published a good overview of potential advantages and challenges of such an approach [4]. This approach may help to obtain a greater sample size, greater follow-up or to include treatments not evaluated in RCT. Another advantages, in a disconnected network, or a network with very unbalanced geometry, i.e. would be to reinforce a weak comparison or reconnect a disconnected network [35]. This appealing method need to satisfy the transitivity assumption and consider the additional uncertainty in treatment effect estimation.

Consistency may be an issue in presence of interaction and treatment modifiers varying across comparison. Researchers relying only on AD may observe an inconsistency if modifier distributions are not equal across comparisons. In our simple network, all data were simulated under the consistency hypothesis, but one-step IPD models could be reformulated (by adding a new term) to detect inconsistency [36,37]. When inconsistency is detected, adjusting for the confounders and/or adding an interaction term may mitigate the issue [11,12,15]. In a recent work, Donegan et al. demonstrated that this problem is more subtle, as it involves both the intercept and slope of the covariate to TE relation [13]. One-step IPD-based models allow flexible handling of this situation with comparison-specific intercept and interaction.

Several strengths of our study should be highlighted. Simulations were done according to 48 scenarios representing a wide set of possibilities and two types of coding (continuous or qualitative) were used for age. Several statistical models were used, including two IPD one-step propositions. Finally, we observed on our real data application that our findings are not just theoretical but can lead to erroneous conclusions in the real world.

Our study had some limitations. On one hand, we restricted our work to a relatively simple three-node network. Whether the model will perform as well and be able to have low bias in a larger network with more complex interaction pattern is unknown. As models become larger, the computational challenge raises, and multiple modifiers can be present. Further works is needed to better address this situation. Another issue may be the geometry of the network [38] as odd geometry can be harder to investigate. For instance, we observed that the ratio between direct and indirect evidence plays a role in the magnitude of the bias). However, it is the first

simulation work to quantify the bias of a frequentist IPD-based approach in presence of interaction for time-to-event data. Our model was fitted on a balanced network i.e with the same amount of information in each comparison, and even if it was not a large network, there was enough information to regress the full model without convergence issues. Whether this approach will work in an unbalanced network or with fewer patients need further works. Future studies may also confirm our findings when inconsistencies in principal effects or interactions are present.

## 8 Conclusion

We proposed a general IPD-based mixed Poisson model in a frequentist framework for network-meta-analysis in presence or not of a modifier treatment for time-to-event data. This model performed well, even in the presence of a TE modifier with unbalanced distribution, and results were stable. When compared to AD-based models, the one-step IPD was the less biased method in all settings and should, therefore, be considered as the gold standard in NMA.

## Supporting information

**S1 Fig.** Distribution of estimated age effect in the Individual Patients' Data based Poisson's one step models according to the true treatment effect with log hazard ratio = -0.5 and -0.2 (rows) and the three scenarios none, interaction and both (columns) for configuration 1 Sigma ($\sigma$) represents the between-trial heterogeneity for the baseline risk and tau ($\tau$) the between-trial heterogeneity of the treatment effect. Red dashed line represents the true effect of age.None: no interaction, same age distribution; Interaction: interaction in AC, same age distribution; Both: interaction in AC, different age distribution.
(TIF)

**S2 Fig.** Distribution of estimated age effect in the Individual Patients' Data based Poisson's one step models according to the true treatment effect with log hazard ratio = -0.5 and -0.2 (rows) and the three scenarios none, interaction and both (columns) for configuration 2 Sigma ($\sigma$) represents the between-trial heterogeneity for the baseline risk and tau ($\tau$) the between-trial heterogeneity of the treatment effect. Red dashed line represents the true effect of age. None: no interaction, same age distribution; Interaction: interaction in AC, same age distribution; Both: interaction in AC, different age distribution.
(TIF)

**S1 Table.** Simulation results (Bias and Empirical Standard Error) of the A-C and B-C pairwise treatment comparisons in all simulation settings Both configuration are 3 nodes (A-B-C) network, with no closed loop in configuration 1 (no A-B trials) and a closed loop in configuration 2. IPD-Poisson is an Individual Patients' Data model based on Poisson's hierarchical model, AD-Metareg an aggregated-data based metaregression model an AD-Netmeta and aggregated-data model based on contrast. ttt: treatment effects as log(HR) with two possibilities: -0.2 and -0.5, $\sigma$: between-trial heterogeneity of baseline risk 0.01 or 0.1), $\tau$: between-trial heterogeneity of treatment effect 0.01 or 0.1), S: scenario with three possibilities (1: no interaction, same age distribution; 2: interaction in AC, same age distribution; 3: interaction in AC, different age distribution), Param: parameters estimated by the model with age in years as a 4-class categorical variable ($<$55, 55–60, 60–65, $>$65) or as a continuous variable (TEage = 60: marginal effect = log(HR) for a patient of age 60; vTEage+1: the variation in log(HR) for a variation of one year of age), NA: Not applicable.
(PDF)

**S2 Table.** Simulation results of the indirect A-B treatment effect estimation with age as a categorical variable (configuration 1) AD: aggregated values, IPD: individual patient's data, ESE: empirical standard error, ASE: Average Standard Error, TE: treatment effect, σ between trial random effect for baseline risk, τ between-trial random effect for treatment effect.
(PDF)

**S3 Table.** Simulation results of the mixed (direct/indirect) A-B treatment effect estimation with age as a categorical variable (configuration 2) AD: aggregated values, IPD: individual patient's data, ESE: empirical standard error, ASE: Average Standard Error, TE: treatment effect, σ between trial random effect for baseline risk, τ between-trial random effect for treatment effect.
(PDF)

**S4 Table.** Simulation results of the indirect A-B treatment effect estimation (configuration 1) and mixed (direct/indirect) estimation (configuration 2) with age as a continuous variable Configuration 1 represents a 3 arms network with B-C and A-C trials from which we estimate the indirect treatment effect AB, configuration 2 represent a 3 arms network with A-B, A-C and B-C trials from which we estimate the mixed (direct and indirect) treatment effect AB. AD: aggregated values, IPD: individual patient's data, ESE: empirical standard error, ASE: Average Standard Error, TE: treatment effect, σ between trial random effect for baseline risk, τ between-trial random effect for treatment effect, TEage = 60: marginal effect = log(HR) for a patient of age 60; vTEage+1: the variation in log(HR) for a variation of one year of age).
(PDF)

**S1 Text. Trials included in the network.**
(PDF)

## Acknowledgments

We thank the investigators and associated research groups of the Meta-Analysis of Chemotherapy in Head and Neck Cancer/Meta-Analysis of Radiotherapy in Squamous Cell Carcinomas of Head and Neck project and the European Organization for Research and Treatment of Cancer, who agreed to share their data.

## Author Contributions

**Conceptualization:** Matthieu Faron, Pierre Blanchard, Laureen Ribassin-Majed, Stefan Michiels, Gwénaël Le Teuff.

**Formal analysis:** Matthieu Faron.

**Methodology:** Matthieu Faron, Pierre Blanchard, Laureen Ribassin-Majed, Jean-Pierre Pignon, Stefan Michiels, Gwénaël Le Teuff.

**Project administration:** Jean-Pierre Pignon.

**Supervision:** Jean-Pierre Pignon, Stefan Michiels.

**Validation:** Matthieu Faron, Pierre Blanchard, Stefan Michiels.

**Writing – original draft:** Matthieu Faron, Gwénaël Le Teuff.

**Writing – review & editing:** Matthieu Faron, Pierre Blanchard, Laureen Ribassin-Majed, Jean-Pierre Pignon, Stefan Michiels, Gwénaël Le Teuff.

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
