## [Decision Letter · Decision Letter 0]

4 Aug 2021

PONE-D-21-19631

A frequentist one-step model for a simple network meta-analysis of time-to-event data in presence of an effect modifier

PLOS ONE

Dear Dr. Faron,

Thank you for submitting your manuscript to PLOS ONE. After careful consideration, we feel that it has merit but needs some minor revisions. Therefore, we invite you to submit a revised version of the manuscript that addresses the points raised during the review process.

We look forward to receiving your revised manuscript.

Kind regards,

Mona Pathak, PhD

Academic Editor

PLOS ONE

2.Please include captions for your Supporting Information files at the end of your manuscript, and update any in-text citations to match accordingly. Please see our Supporting Information guidelines for more information: http://journals.plos.org/plosone/s/supporting-information.

Additional Editor Comments (if provided):

The manuscript is well planned and drafted. I have few minor comments/suggestion to further improve the manuscript.

1. It is nice to explain with simplest model but practically we deal with multiple treatment network (mostly greater than three). In such a situation, how these can be extended in view of multi-collinearity and multiple interaction. Please discuss under discussion section.

2. For table 3, a footnote specifying configuration 1 as three node NMA model without loop and configuration 2 is model with loop should be specified.

3. In view of computation requirement, author have not simulated for mixed effect Cox proportional hazard. The performance of CoxME is quite close to IPD-Poisson ME1, please discuss.

4. Under Table 5, for mRT vs RT comparison CoxME has relatively lower standard deviation in comparison to Poisson ME1, still CoxME has broader 95% confidence interval than that of Poisson, please explain.

Reviewers' comments:

Reviewer's Responses to Questions

**Comments to the Author**

1. Is the manuscript technically sound, and do the data support the conclusions?

Reviewer #1: Yes

2. Has the statistical analysis been performed appropriately and rigorously? 

Reviewer #1: Yes

3. Have the authors made all data underlying the findings in their manuscript fully available?

Reviewer #1: Yes

4. Is the manuscript presented in an intelligible fashion and written in standard English?

Reviewer #1: Yes

5. Review Comments to the Author

Reviewer #1: Thank you for the opportunity to review the manuscript on a frequentist one-step model for a simple network meta-analysis of time-to-event data in presence of an effect modifier. The article is well-written and scientifically sound, however, I have some concerns that the authors might want to clarify in their manuscript:

1) Line 6 (introduction) you state that “network is created whose nodes are therapeutic options”...nevertheless, this concept is broader and ‘nodes’ in a network meta-analysis may refer to other technologies (e.g. diagnostic approaches, health systems, preventive measures…). Please consider reformulating this concept

2) Line 12: Likewise, the identification of “all trials” could also refer to other sources of information (e.g. observational studies such as cohorts or case control studies). Please reformulate this part of the introduction to avoid misleading interpretation.

3) Line 35: add further concepts on transitivity and consistency as these are importante assumptions of NMA that should be attained.

4) Why meta-analyses of chemotherapy in Head and Neck Cancer (MACH-NC) and Radiotherapy in Carcinomas of Head and 50 Neck (MARCH) were used for 51 illustration? Better justify this selection in the introduction

5) Better explain the reasons for selecting the multilevel Cox proportional hazard model (CoxME) only for the MARCH/MACH-NC NMA application

6) How this type of model would behave in the presence of a complex network? Why this was not explored in the research?

7) What would be the procedure in case of confounders or imbalanced information?

8) Why a frequentist framework was selected?

9) How network geometry may influence on the results? This should be better discussed in your study. See for instance the paper PLoS One. 2019 Feb 20;14(2):e0212650.

6. PLOS authors have the option to publish the peer review history of their article (what does this mean?). If published, this will include your full peer review and any attached files.

Reviewer #1: **Yes: **Fernanda S. Tonin

---

## [Author Response · Author response to Decision Letter 0]

17 Sep 2021

First, we would like to thank the editors and reviewers for their time and precious comments.

Journal requirement

Manuscript

The manuscript has been modified in order to exactly meet the PLOS One template. 

Authors title and affiliations have been modified to meets PLOS ONE requirement. 

The reference list has been reviewed and is complete and correct. To the best of our knowledge we are not citing any retracted manuscript.

Supporting information

The supporting information has been modified to fulfill requirements. All captions for supplementary table and figures have been moved to the end of the manuscript.

The former supporting file has been divided in:

• 1 supporting text (S1_Text)

• 2 supporting figures (S1_Fig and S2_Fig)

• 4 supporting tables (S1_Table, S2_Table, S3_Table and S4_Table)

All files are PDF and have been named as requested.

Comments from Editor

We thank the editor for their suggestions to improve the manuscript.

Point 1: “It is nice to explain with simplest model but practically we deal with multiple treatment network (mostly greater than three). In such a situation, how these can be extended in view of multi-collinearity and multiple interaction. Please discuss under discussion section.”

We here developed a frequentist framework limited to a 3-treatments network meta-analysis for time-to-event individual data in the presence of modifier and evaluated its performance through a large simulation study. One perspective is to evaluate this approach in more complex networks with multiple treatments and multiple interactions as we agree with the reviewer that multiple pairwise comparisons (more than 3) are commonly performed in practice. Technically our approach can be extended in more complex networks with different structures and multiple interactions, but this was not the scope of this work. With more complex networks, the parameters number to be estimated increases and extensions must consider the parameters estimation issues. One option we discussed is the use of penalized regression methods. 

Consequently, the discussion section has been modified to discuss this point:

“As models become larger, the computational challenge raises, and multiple modifiers can be present. Further works is needed to better address this situation. Another issue may be the geometry of the network[35] as odd geometry can be harder to investigate. For instance, we observed that the ratio between direct and indirect evidence plays a role in the magnitude of the bias).”

Point 2: “For table 3, a footnote specifying configuration 1 as three node NMA model without loop and configuration 2 is model with loop should be specified.”

The table 3 footnote had been modified to precise that configuration 1 is a 3-nodes model without a closed loop and configuration is 2 the same model with a closed loop.

Point 3: “In view of computation requirement, author have not simulated for mixed effect Cox proportional hazard. The performance of CoxME is quite close to IPD-Poisson ME1, please discuss.”

The Cox and Poisson models have identical or very similar results depending on the way the time is partitioned when using Poisson model. Indeed, if the data are splitted at every time of event, the likelihood functions of these two models are identical and give identical inferences for parameter estimations. Because baseline risks in clinical trial does not vary so rapidly over time in our experience, Poisson’s model with one-year interval gave results practically identical to those of the Cox Model in oncology trials in the adjuvant, locally advanced or advanced setting. In a simple model with a small dataset the optimization process of the likelihood function of these 2 models are not computationally intensive. Yet, in large datasets or with multiple interactions and random-effects, Poisson’s model converged faster than Cox model. For example, in our application to the MARCH/MACH-NC NMA, the time ratio was 1:4: Poisson’s model converged in 100 seconds when Cox model took 400 seconds. This is also the interest of our approach to propose a Poisson model for network meta-analysis with time-to-event data. Our simulations present very varied scenarios, and 48,000 models were simulated. Simulating these models took several days on the 80 cores / 256 GB memory server of the lab. Using a Cox-model would have taken more than a week. Regarding the execution time, we thus decided to use the Poisson's model for the large simulation study but both models were used for the real network meta-analysis. The following phrases were added to the discussion:

“For large simulation, using the Cox model may become very computationally intensive. In our simulation, despite the use of parallel computing and an 80-cores server, it would have taken more than a week. For this reason, we only used Poisson model in the simulation study.”

And a phrase in the method section was also modified:

“Because of too long computation time when calculating a lot of models, this model was not used in the simulation study […]”

"Because of the too long computation time of the optimization process of a Cox model with random effects, it was not used for practical reason in the large simulation study". 

Point 4: “Under Table 5, for mRT vs RT comparison CoxME has relatively lower standard deviation in comparison to Poisson ME1, still CoxME has broader 95% confidence interval than that of Poisson, please explain.”

In Table 5, the standard error of CoxME and PoissonME1 for the mRT vs RT comparison is 0.045 and 0.048, respectively. The estimate of standard error of PoissonME1 is slightly higher to that of CoxME. The confidence interval of CoxME and PoissonME1 was [-0.211, -0.035] (width=0.176) and [-0.215, -0.026] (width=0.189), respectively. 

Comments from reviewer 1

We are grateful to reviewer 1 comments and suggestion about our work.

Point 1) Line 6 (introduction) you state that “network is created whose nodes are therapeutic options” ...nevertheless, this concept is broader and ‘nodes’ in a network meta-analysis may refer to other technologies (e.g. diagnostic approaches, health systems, preventive measures…). Please consider reformulating this concept

Thanks for the suggestions. We added this phrase to better acknowledge that any interventions can be analyzed with this methodology:

“NMA can also be applied to interventions such as diagnostic or preventive measures.” 

Point 2) Line 12: Likewise, the identification of “all trials” could also refer to other sources of information (e.g. observational studies such as cohorts or case control studies). Please reformulate this part of the introduction to avoid misleading interpretation.

Thanks again for the suggestion. We focused our work on randomized controlled trial because this type of study gives the highest level of evidence. At the beginning of the paragraph, we replaced the term “trial” with “study” and then add this phrase:

“While, any type of study can be used (observational, case-control, cohort and RCT), the quality of the MA/NMA relies on the quality of the underlying studies. We focused on RCTs as they are considered to provide the the highest level of evidence for treatment effect evaluation in the context of evidence-based medicine and because the incorporation of non RCT in NMA would address new challenges not considered here.”

In the discussion we also add that an NMA can also be a mixture of trial with other type of study, notably when information is lacking in some comparison. The following phrase was added to the discussion:

“Others have proposed to combine RCT and non RCT study in the network. Cameron et al. published a good overview of potential advantages and challenges of such an approach.[4] This approach may help to obtain a greater sample size, greater follow-up or to include treatments not evaluated in RCT. Another advantages, in a disconnected network, or a network with very unbalanced geometry,, i.e. would be to re-inforce a weak comparison or reconnect a disconnected network. [34]This appealing method need to satisfy the transitivity assumption and consider the additional uncertainty in treatment effect estimation.”

Point 3) Line 35: add further concepts on transitivity and consistency as these are important assumptions of NMA that should be attained.

Thanks for the suggestion. We totally agree that these concepts are important. In fact, a whole paragraph was present on a previous version of the manuscript but then discarded to save place. The following phrases has been added:

“Transitivity corresponds to the fact that all patients of the network could have been randomized to any treatment options or in other words that there are no systematic differences between the comparisons. Transitivity assumption does not hold when an effect modifier has not a similar distribution across comparisons. Consistency, which can be considered as the manifestation of transitivity in the data, means that for a particular treatment comparison where both direct and indirect evidence are available, they provide similar estimations.”

Point 4) Why meta-analyses of chemotherapy in Head and Neck Cancer (MACH-NC) and Radiotherapy in Carcinomas of Head and 50 Neck (MARCH) were used for 51 illustration? Better justify this selection in the introduction

In fact, this NMA was the primary reason why we did this study. While investigating the results of the MARCH MA we observed the interaction between the treatment effect and age. When we used this MA in the NMA we wondered what bias this interaction can produce if ages were distributed differently. Besides we don’t have so many IPD-NMA available with the presence of a treatment effect modifier. The following precision was added to the introduction:

“This NMA was chosen because of a previously observed interaction between age and treatment effect as published in the main papers of MARCH/MACH-NC.” 

Point 5) Better explain the reasons for selecting the multilevel Cox proportional hazard model (CoxME) only for the MARCH/MACH-NC NMA application

Thanks for this suggestion which has also been made by the editor. The reason was purely for computational time. Why we do know that Cox and Poisson hierarchical model give very close results, the computation time is not the same. The differences are getting bigger as the model became more complex. For instance, the calculation of the single model of MARCH/MACH-NC, by Poisson’s model took 100 seconds, and Cox’s model, four time as much. We simulated 48,000 different datasets and calculating only Poisson’s model took several days on the 80 cores / 256 GB memory server of the lab. Doing Cox’s model would have taken more than a week, and we felt that too long for the potential benefits it added. The discussion has been modified to better precise this fact:

“The Cox model however can become very computer intensive as compared to the Poisson implementation, and for this reason, we only used the Poisson model in the simulation.”

And a phrase in the method section was also modified:

“In order to reduce computation time in the simulation study,”

Point 6) How this type of model would behave in the presence of a complex network? Why this was not explored in the research?

Thanks for this important question. We currently don’t know how the model will behave in a complex network. Our idea was to focus first on the simplest possible network (3-nodes) and to expand to more complex network in a second study. As networks becomes larger, the imbalance in the distribution of the modifier or the interaction can come from more and more comparison and thus become hard to interpret. We felt that, for the first study, a simple network will be easier to understand. We modify the discussion section to better emphasize on this fact:

“As models become larger, the computational challenge raises, and multiple modifiers can be present. Further works is needed to better address this situation. Another issue may be the geometry of the network[35] as odd geometry can be harder to investigate. For instance, we observed that the ratio between direct and indirect evidence plays a role in the magnitude of the bias).”

Point 7) What would be the procedure in case of confounders or imbalanced information?

Thanks again for this excellent question. 

We have added a sentence to the introduction with reference to papers that document the causal inference methods used for this type of question 

“For NMA based on non-randomized studies, IPD may also be used to reduce confusion bias.”

Point 8) Why a frequentist framework was selected?

While we appreciate the advantages of Bayesian analysis, our team has more frequently used the frequentist paradigm. Our review of the available literature showed that while some team used Bayesian analysis for time to event data (particularly the MRC-London team) very few used a frequentist approach. We therefore decided to focus on this approach. A paragraph on the discussion acknowledges this point but we can expand more if you feel it’s necessary:

“Correctly estimating networks may be computationally challenging. Both Bayesian and frequentist estimation tools are available for AD-NMA and considered to deliver similar results. For IPD, most teams use either frequentist two-step or Bayesian one-step IPD. The former use IPD data to derive trial-specific log hazard ratios adjusted for other variables which are then combined. The latter allows more complex models, sometimes at the price of long calculation times. In practice, very few one-step frequentist studies have been published, especially for time-to-event outcomes, as specifying the model may be challenging and models with multiple random effects and interactions may never converge.”

Point 9) How network geometry may influence on the results? This should be better discussed in your study. See for instance the paper PLoS One. 2019 Feb 20;14(2):e0212650.

Thanks for your suggestion and the very interesting paper. As discussed in point 6 a more complex network will be more difficult, and the difficulty will raise if the geometry of the network is unfavorable. Understating how our model performs in such a situation will need another work. Yet, even in our simple 3-nodes network, as you stated in your paper, two networks with the same number of nodes/edges and comparison could correspond to different networks. In the context of our study a major point would be how the available evidence are distributed between the direct and indirect evidence. A great care should be taken when some comparison relies only on indirect evidence or “weak edges”. The dispersion value of the thickness can also alert in case of imbalance. As we observed, adding direct evidence was able to mitigate the bias and network with a lot of direct information on the comparison of interest are less prone to interaction problems in other edges. The following two sentences have been added with a reference to the paper you suggested

“Another issue might be the geometry of the network [34] as odd geometry may be harder to investigate. For instance, we observed that the ratio between direct and indirect evidence plays a role in the magnitude of the bias.”

---

## [Editor Report · Decision Letter 1]

13 Oct 2021

A frequentist one-step model for a simple network meta-analysis of time-to-event data in presence of an effect modifier

PONE-D-21-19631R1

Dear Dr. Faron,

We’re pleased to inform you that your manuscript has been judged scientifically suitable for publication and will be formally accepted for publication once it meets all outstanding technical requirements.

Kind regards,

Mona Pathak, PhD

Academic Editor

PLOS ONE

---

## [Editor Report · Acceptance letter]

22 Oct 2021

PONE-D-21-19631R1 

A frequentist one-step model for a simple network meta-analysis of time-to-event data in presence of an effect modifier 

Dear Dr. Faron:

I'm pleased to inform you that your manuscript has been deemed suitable for publication in PLOS ONE. Congratulations! Your manuscript is now with our production department. 

Kind regards, 

on behalf of

Dr. Mona Pathak 

Academic Editor

PLOS ONE